# Serological Responses of Guinea Pigs and Heifers to Eight Different BoAHV-1 Vaccine Formulations

**DOI:** 10.3390/vaccines12060615

**Published:** 2024-06-04

**Authors:** Luana Camargo, Yasmin Vieira Franklin, Gustavo Feliciano Resende da Silva, Janaína Ferreira Santos, Viviana Gladys Parreño, Andrés Wigdorovitz, Viviani Gomes

**Affiliations:** 1Department of Medical Clinic, Universidade de São Paulo (USP), São Paulo 05508270, SP, Brazil; luanamcamargo@usp.br (L.C.); yasmin.franklin@usp.br (Y.V.F.); janaina.ferreira@usp.br (J.F.S.); 2Department of Animal Production, Universidade Federal de Goiânia (UFG), Goiânia 74690900, GO, Brazil; gustavofeliciano@hotmail.com; 3Institute of Virology, ICCVyA-INTA, Cdad. Autónoma de Buenos Aires C1033, Argentina; parreno.viviana@inta.gob.ar (V.G.P.); wigdorovitz.andres@inta.gob.ar (A.W.)

**Keywords:** BoAHV-1, vaccination, dairy cattle, vaccine, antibody

## Abstract

Bovine alphaherpesvirus 1 (BoAHV-1) infection affects the production and reproductive performance of dairy and beef livestock, resulting in considerable economic losses. In addition to biosecurity measures, vaccination programs are effective strategies for controlling and preventing BoAHV-1 infection and transmission. We evaluated the serological immune response against BoAHV-1 induced by eight different formulations of commercial vaccines: three modified live vaccines and five killed vaccines containing BoAHV type 1 or types 1 and 5. In the first experiment, 50 BoAHV-1-seronegative guinea pigs were assigned to eight groups; each individual in the treatment groups received two doses (one-fifth of the bovine dose). The second experiment was conducted using 29 crossbred Holstein × Gir heifers in four groups of six to nine animals each. The serological immune response against BoAHV-1 was measured using virus neutralization and enzyme-linked immunosorbent assays to measure the total IgG against BoAHV. We evaluated the effects of the vaccine, time, and interaction of the vaccine and time on neutralizing antibodies against BoAHV-1. Killed vaccines produced low levels of antibodies against BoAHV-1, whereas modified live vaccines produced high levels of antibodies capable of providing neutralizing titers in the vaccinated animals, with the thermosensitive modified live vaccine showing the highest levels of antibodies.

## 1. Introduction

Bovine alphaherpesvirus 1 (BoAHV-1) belongs to the *Alphaherpesviridae* subfamily, the genus *Varicellovirus* and the *Varicellovirus bovinealpha* species [1]. BoAHV-1 is classified into three genetic subgroups: BoAHV-1.1, which is associated with respiratory and reproductive diseases and occurs in North America, South America, and Europe; BoAHV-1.2a, which is associated with reproductive, respiratory, and genital diseases in particular (infectious pustular vulvovaginitis and infectious balanoposthitis) and is prevalent in Brazil; and BoAHV-1.2b, associated with localized respiratory and genital diseases and prevalent in Europe, the US, and Australia [2]. 

Bovine alphaherpesvirus 5 (BoAHV-5) causes meningoencephalitis in young cattle and shares close antigenic and genetic similarities with BoAHV-1. Both viruses follow similar pathways in their progression: these viruses (1) invade epithelial cells at the entry point and (2) establish a latent state within sensory nerve ganglia, specifically the trigeminal ganglia. However, their abilities to invade the nervous system and cause neurological symptoms differ [3]. A study in South America showed that 82.8% of the BoAHV-positive population was (latently) infected with BoAHV-1, 93.1% with BoAHV-5, and 75.9% with both BoAHV-1 and BoAHV-5. This marks the first documented instance of such a high frequency of co-infection with BoAHV-1 and BoAHV-5 in bovines [4].

BoAHV-1 causes damage to beef and dairy farming worldwide. The direct costs of BoAHV-1 infection in the UK farming industry are estimated to be GBP 4 million per year [5]. Milk production in BoAHV-1 serum-positive cows is reduced by 1000 kg milk per year, which has severe economic consequences for producers, in addition to negative impacts on the animals’ health and well-being [6].

Herpesviruses use the host cell nucleus for viral DNA synthesis, encoding a large number of proteins, and can cause latent infections during which the genome is retained in its episomal form to generate little or no gene expression. BoAHV-1 persists in herds because it is silently carried, exhibiting a characteristic subclinical disease status. This factor means that it is identified with difficulty by farm workers without serological tests, allowing the virus to go unnoticed and persist silently in herds, ensuring its perpetuated spread [7]. 

Like most viral infections, immune responses against BoAHV-1 can be divided into cellular and humoral responses [8]. The primary immune response stimulates humoral and cell-mediated responses in the host. The cellular immune response to BoAHV-1 can initially be detected at 5 days after infection, reaching a peak between 8 and 10 days. Neutralizing antibodies, mainly IgM and IgG, can be detected approximately 10 days after infection. Neutralizing IgA may occur in the nasal and genital secretions of infected animals. Although neutralizing antibodies play important roles in the immune response, animal recovery from BoAHV-1 challenge predominantly depends on cell-mediated immunity [9]. 

Modified live vaccines (MLVs) and killed vaccines (KLVs), administered intranasally or parenterally, have been used to control the spread of BoAHV-1 in herds and to reduce the clinical disease severity and economic losses associated with reproductive failure and miscarriage [10,11]. KLV contains inert viral particles that prevent virus replication in the host; however, this immune predominantly occurs as Th2 cells mediated by B lymphocytes (humoral), which produce neutralizing antibodies. KLVs cannot generate stimuli for cellular immunity, which is the main host defense mechanism against viral infections [12]. MLVs have been used in control programs because of their strong antibody response, long duration of immunity, few doses needed per animal, and low costs [13]. Although MLVs appear to generate a strong protective response, a literature review concluded that some live vaccines against BoAHV-1 have severe effects on ovarian function [10], which may culminate in fetal losses, raising considerable doubts regarding the “safety/exemption” of the parenteral administration of anti-BoAHV-1 vaccines containing a modified live virus. Therefore, MLVs are not recommended for vaccinating pregnant animals [13].

Guinea pigs are established experimental models for testing the efficacy of bovine vaccines against BoAHV-1, as bovine trials are complex, costly, and time-consuming, particularly in countries where BoAHV-1 is endemic [14]. A guinea pig model was previously established to estimate the potency and efficacy of KLV against BoAHV-1 for cattle [15]; the study revealed a strong association between the model and target species for antibody titers measured using virus neutralization (VN) and an enzyme-linked immunosorbent assay (ELISA).

Considering the importance of efficient vaccines that can generate a protective and lasting immune response for the prevention of reproductive diseases, this study was conducted to evaluate the serological response induced by eight different vaccine formulations available on the Brazilian market in guinea pigs and Holstein × Gir heifers. 

## 2. Materials and Methods

The guinea pig experiment was conducted from September to December 2021 and the heifer experiment was conducted from December 2021 to February 2022, followed by antibody measurements. Animal procedures were conducted in strict accordance with the norms of the Brazilian Committee for Animal Experimentation (law n. 6638 of 8 May 1979) and were approved by the Ethics Committee on the Use of Animals of the School of Veterinary Medicine and Animal Science of the University of São Paulo (CEUA/FMVZ, protocol number 7782291020). 

### 2.1. Vaccine Formulation

Commercial vaccines were selected according to their formulation and registration on the website of the Ministério da Agricultura, Pecuária e Abastecimento, Brazil. Vaccines with BoAHV-1 genotypes associated or not with other antigens were evaluated (Table A1).

### 2.2. Vaccination Schemes and Sampling Times

The guinea pig experiment was conducted according to recommendations submitted to the Committee of the Americas on Veterinary Medicines, resolution 598/2012 of the Official Gazette of 4 December 2012. The use of the “Guinea Pig INTA” model is also specified in article 38, which states that “potency controls will be carried out in guinea pigs”, with tests based on two immunization doses in an animal at an interval of 21 days. At 30 days, a serum sample should be taken for analysis; this was developed by INTA and is detailed in the technical annex of the resolution [13].

Fifty male Hartley guinea pigs, weighing approximately 400–450 g and aged more than 30 days, were randomly assigned to seven experimental groups (vaccines A–G). Each group contained six animals each, except for groups E and C, which comprised seven animals each, and the control group with six animals. The guinea pigs were allowed to acclimatize for 7 days before the initial vaccination. The animals were vaccinated subcutaneously with two doses of commercial formulations at an interval of 21 days (Figure 1), with a volume corresponding to one-fifth of the bovine dose. The animals were observed for a minimum of 30 days, and serum samples were collected at T0 and nine days after revaccination (T30) to determine the antibody titers using a VN assay and an indirect ELISA. In the second experiment, 29 seronegative crossbred Holstein x Gir heifers aged 7–12 months were used. Heifers were randomly assigned to four treatment groups, i.e., vaccination with MLV B (*n* = 7), with thermosensitive vaccine D (*n* = 7), and with MLV H (*n* = 9), in addition to an unvaccinated control group (*n* = 6). Serum samples were collected on day 0 (during the first vaccination) and 60 days thereafter (day 60; Figure 1). Vaccination was performed according to the manufacturers’ instructions.

#### 2.2.1. Virus Neutralization

BoAHV-1 (Cooper) was used in the VN test for guinea pig and bovine samples. Amplification procedures, viral titration, and serological tests were performed using bovine kidney cells of the MadinDarby bovine kidney line (MDBK), which is free of pestivirus. The cells were maintained in minimal essential medium (Vitrocell, Nova Campinas, São Paulo, Brazil), supplemented with penicillin (10,000 IU/mL), streptomycin (10 mg/mL), ciprofloxacin (10 mg/mL), and amphotericin B (250 µg/mL). MDBK cells were cultured in 10% equine serum. VN assays were performed in 96-well plates containing increasing dilutions of serum, starting at a ratio of 1:4. The samples were incubated with 100–200 TCID50 of BoAHV-1 for 2 h. A suspension of MDBK cells was added, and the plates were incubated at 37 °C with 5% CO_2_. Neutralizing titers were determined according to the presence or absence of cytopathic effects in infected cells after 96 h of incubation. Antibody titers were considered as the reciprocal of the highest serum dilution that prevented the production of a cytopathic effect.

#### 2.2.2. ELISA to Quantify Total Antibody to BoAHV-1 in Bovine

Briefly, polystyrene microtiter ELISA plates (Immulon 1B, Dynatech Laboratories, Fairfax, VA, USA) were coated with 50 μL of positive (concentrated and semi-purified BoAHV-1 reference strain Los Angeles containing 109 DICT50/mL) or negative antigen (non-infected MDBK cells) in carbonate buffer (pH 9.6) and incubated for 12 h at 4 °C. The plates were blocked with 10% ovalbumin in PBS-Tween 20 0.05%, washed twice, and stored at −20 °C until use. Samples were tested at 1:4, 1:10, and 1:40 dilutions for herd surveys and in six serial four-fold dilutions starting at a 1:40 dilution for vaccine potency testing. For standardization, as described for VN, samples from immunized animals inoculated with sub-potent calibration vaccines (containing 105 TCID50/per dose) that were negative at a 1:40 dilution were re-tested starting at a 1:4 dilution. A peroxidase-labeled affinity-purified goat anti-bovine IgG (H + L) (Kirkegaard & Perry Laboratories, Gaithersburg, MD, USA) in a 1:4000 dilution was used as the conjugate. After each incubation, the plates were washed four times. The reaction was developed using hydrogen peroxide/2,2′-azino-bis 3-ethylbenzthiazoline-6 sulfonic acid in citrate buffer (pH 5) as the substrate/chromogen system. The antibody titer of each sample was expressed as the log10 of the reciprocal of the highest serum dilution with a corrected optical density (OD405 in the positive coated wells minus OD405 in the negative coated wells) greater than the cut-off value of the assay [15].

#### 2.2.3. ELISA to Quantify Total Antibody to BoAHV-1 in Guinea Pig

The ELISA was adapted to detect guinea pig antibodies using the same procedure described for bovine samples, and a peroxidase-labeled affinity purified goat anti-guinea pig IgG (H + L) (Kirkegaard & Perry Laboratories) was used as the conjugate in a 1:4000 working dilution. As an alternative model for vaccine potency testing, samples of immunized guinea pigs were tested at six serial four-fold dilutions starting at a 1:40 dilution [15].

### 2.3. Statistical Analyses

A statistical model was used for antibodies from the bovine model and guinea pig model. Statistical analyses were performed using SAS statistical software (version 9.4, SAS Institute, Cary, NC, USA). The titers of antibodies against BoAHV-1 were evaluated in relation to the Gaussian distribution using the command Guided Data Analysis, and the normal distribution of the data was tested using the Shapiro–Wilk or W test (*p* ≥ 0.05). All antibody titers were classified as normal variables, and real-value transformation was performed. The results are shown as the means ± standard error. In the guinea pig experiment, a model was fitted with the fixed effects of treatments (vaccines A–G and control) and time (days 0 and 30), as well as the interaction effect of treatment and time, using the MIXED procedure (PROC-mixed, SAS) with a least significant difference (LSD) post hoc test. The models were tested according to covariance structures using the Akaike information criterion. In the heifer experiment, a model was fitted with the fixed effects of treatments (vaccines B, D, G, and control) and time (days 0, 30, and 60), as well as the interaction effect of treatment and time using the MIXED procedure (PROC-mixed, SAS), with an LSD post hoc test. The models were tested according to covariance structures using the Akaike information criterion. Statistical significance is reported at *p* ≤ 0.05. 

## 3. Results

### 3.1. Virus Neutralization

The treatment, time, and their interaction had significant effects on neutralizing antibody titers in the guinea pig experiment (*p* < 0.001, each; Table 1). Comparisons of the antibody titers between the vaccine formulations on days 0 and 30 of the guinea pig experiment are shown in Table 2. The basal neutralizing antibody titers (at day 0) were 4.00 ± 0.00 in all vaccine groups. The controls remained seronegative for BoAHV-1 throughout the experiment. On day 30, only vaccine formulations B and D produced strong serological responses against BoAHV-1 (36.00 ± 9.63 and 53.33 ± 6.74, respectively). Thus, vaccines B and D were used in the heifer experiment. Five of the six guinea pigs died after the first dose of vaccine A and two of the six guinea pigs died after the first dose of vaccine C.

Comparisons of the neutralizing antibody titers between vaccine formulations on days 0 and 60 of the heifer experiment are also shown in Table 2. The humoral immune response that developed after the first and second doses appeared to be better than that in the guinea pig experiment. For BoAHV-1, all animals had serum-neutralizing titers of less than 4.00 on day 0. Vaccine D (210.28 ± 57.09) induced the highest antibody titers, followed by vaccines H (74.66 ± 14.11) and B (36.57 ± 7.58). 

In Figure 2, it is possible to verify the dispersion of the antibody levels of each animal according to the vaccine used. The distribution (%) of antibody titers against BoAHV-1 induced by commercial vaccines in heifers is shown in Figure 3 and Table A1 in Appendix A. Antibodies against BoAHV-1 (≥32) on day 60 were detected in four of the seven (57.2%) individuals in the vaccine B group, seven of the seven (100%) in the vaccine D group, and nine of the nine (100%) in the vaccine H group.

### 3.2. ELISA

The treatment group, time, and their interaction significantly affected specific antibody titers against BoAHV-1 (*p* < 0.001; Table 3). 

A comparison of the antibody titers on days 0 and 30 in the guinea pig experiment is shown in Table 4. The basal antibody titers were 0.00 ± 0.00 in all experimental vaccine groups. The control group remained seronegative throughout the experiment. In the guinea pig experiment, vaccines B and D showed an increase of (1066.67 ± 486.47) and (1386.67 ± 533.33), respectively. Vaccines B and D were used in the heifer experiment. A comparison of the vaccination results in the heifers is shown in Table 4 also. Antibodies were detected on day 60 in the vaccine B (1188.57 ± 354.10), vaccine D (2011.43 ± 354.10), and vaccine H (1449.00 ± 357.77) groups.

## 4. Discussion

We evaluated the immunogenicity of MLVs and KLVs against BoAHV-1 by measuring the serological responses in guinea pigs and Holstein × Gir heifers.

Guinea pigs are a reliable indicator of vaccine immunogenicity and protection against BoAHV-1 in the natural host [13]. Although BoAHV-1 is specific to ruminants, guinea pigs can be used to predict how the immune system will respond to vaccination. A statistical analysis was conducted to categorize the vaccine efficacy, and the agreement between the laboratory animal model and the natural host was verified through an experimental challenge of BoAHV-1. The results showed that the antibody levels in vaccinated guinea pigs can be used as a reliable tool to predict vaccine efficacy in cattle [13].

According to Pospísil et al. [16], the minimum antibody titers required to protect animals from BoAHV-1 challenge are ≥16 or 32; however, higher neutralizing antibody titers likely provide better protection. In the present study, the serological response to BoAHV-1 was analyzed by comparing MLVs and KLVs in guinea pigs and Holstein × Gir heifers. After the first dose of vaccine A and vaccine C, a large proportion of the guinea pigs in these groups died. The animals presented with hemorrhage and pulmonary edema. Vaccine A and C contained the inactivated suspension of antigens from BoAHV. But these formulations contain high levels of Gram-negative bacteria and thus are not safe for use in guinea pig species. The volume of the dose administered to the guinea pigs was 1/5 of the volume of the dose given to bovines [17]. 

In the first study, KLV E, F, and G did not induce satisfactory levels of antibodies to protect against BoAHV-1 infection. In contrast, the two MLVs evaluated in the guinea pig experiment (vaccines B and D) produced detectable levels of neutralizing antibodies capable of protecting the natural hosts from natural infection. The antibody titer results of the Holstein × Gir control heifers were similar to those in guinea pigs; however, the intensity of the developed serological response appeared to be stronger in heifers. In the guinea pig experiment, only vaccines B and D induced a notable increase in antibody titers. Vaccine D induced the highest antibody titers, followed by vaccine B, which produced lower titers on day 30. In heifers, formulations B, D, and H induced the highest antibody titers. Vaccine D is a lyophilized preparation of chemically altered thermosensitive BoAHV strains. The H vaccine was composed of modified live samples of the infectious bovine rhinotracheitis virus passage strain C-13. Vaccine B contained live gE/TK-double-deletion BoAHV-1, strain CEDDEL: 106.3–107.3 CCID50. 

Similar results were reported by De Brun et al. in Uruguay [11]. The authors examined the effectiveness of four inactivated vaccines that are commonly available in Uruguay against BoAHV-1 by measuring the levels of neutralizing antibodies and the subclasses of specific immunoglobulins induced by the vaccines. Similar to our results, only one inactivated vaccine reached the minimum titer (1:8) considered as adequate for animal protection. In contrast, IgG2 titers were high for all inactivated vaccines, indicating a Th1 response profile adequate for an antiviral response. These results [11] and those of our study suggest that the level of neutralizing antibodies alone may not be an adequate parameter for determining protection against BoAHV-1.

We used the BoAHV Cooper strain to perform VN tests. In contrast to the results obtained for inactivated vaccines in guinea pigs, the MLVs B, D, and H induced a large increase in antibody levels over time, suggesting immunological protection in the case of a real infection by BoAHV-1; however, the protective response is not based exclusively on the production of detectable levels of neutralizing antibodies. The antibody titers produced by vaccines B, D, and H were 36.57 ± 7.58, 210.28 ± 57.09, and 74.66 ± 14.11, respectively. All antibody titers increased over time; however, vaccine D (a modified thermosensitive live vaccine) showed a greater increase in antibody levels. Silva et al. [18] employed calves aged 10 to 14 months to assess the immunogenicity of six commercially available vaccines containing inactivated BoHV-1 antigens: one from Brazil (BR), one from North America (US), two from Uruguay (UR1 and UR2), and two from Argentina (ARG1 and ARG2). The US vaccine demonstrated an 87.5% success rate in achieving minimum titers (≥16 or 32) in the animals. Baccili et al. [19] analyzed the vaccine response against BoAHV-1, comparing MLVs and KLVs. Similar to our results, the best protective response was observed in heifers vaccinated with a live modified thermosensitive virus. Evidence suggests that MLVs generate a broader and more effective immune response against BoAHV-1 than inactivated vaccines [20,21]. The variability in the effectiveness of BoHV-1 vaccines across South American countries is intriguing. These variations can be explained by viral variants, environmental conditions, and the history of exposure and immunity of the target herd [11,18,19,20,21]. 

Walz et al. also demonstrated that the annual revaccination of pregnant cows with thermosensitive vaccines is effective in preventing abortion and fetal infection by BoAHV-1 after two pre-breeding doses of MLV vaccines. Annual revaccination with the thermosensitive vaccine provided greater protection against fetal viral infection [22]. For calves, the vaccination frequency is critical to ensuring a good immune response. Typically, vaccination programs stipulate that calves should receive vaccinations as per protocols within the initial two months of life. Excessive vaccination in young calves may induce antigen-specific tolerance, characterized by T cell suppression and T and B cell deletion, or trigger autoimmune responses [23].

Perry et al. [24] demonstrated that vaccination with an MLV before the start of the breeding season was associated with decreased conception rates for fixed-time artificial insemination, regardless of when vaccination occurred. The exact mechanism underlying these reductions in conception rates is not known; however, these points should be considered before choosing the optimal commercial BoAHV-1 vaccine.

The selection of optimal vaccination strategies involves the careful consideration of safety and efficacy [24]. A recent study using multivalent BoAHV-1 and BVDV vaccine regimens with two doses of MLVs in heifers, followed by an inactivated vaccine, demonstrated superior protection against BoAHV-1 abortion compared with the use of only multivalent MLVs [24]. Farmers often opt to administer a different vaccine brand for the second dose, a practice believed, anecdotally, to elicit a superior immune response compared to using the same brand for both vaccinations [22]. Vaccination with a thermosensitive vaccine may be a better option for pregnant cows because of its higher level of protection against BoAHV-1 and the potential safety concerns associated with the use of MLV in pregnant cows [24]. According to Chase et al. [10], the ideal reproductive multivalent viral vaccine may contain inactivated BoAHV-1 with the BVDV MLV, but current vaccines do not contain this antigen combination. 

We measured only the serological responses of the animals after vaccination, which may bias the results depending on the vaccine formulation used. To more comprehensively assess vaccine efficacy, further research is needed to measure the cellular immune responses in guinea pigs and subsequently in heifers. In theory, vaccines incorporating ISCOM complexes are believed to have a heightened capacity to stimulate Th1 cellular immune responses. Effective vaccination should trigger both humoral and cellular immune responses to safeguard cattle against viral pathogens. 

## 5. Conclusions

The KLV produced low levels of antibodies against BoAHV-1, and the MLV produced high levels of antibodies capable of providing protective titers in vaccinated animals, with the thermosensitive MLV leading to the highest antibody levels; however, further studies are needed to understand the mechanisms of viral reactivation and safety in pregnant cows. Additionally, the accuracy of measuring antibody titers to assess vaccine efficacy against BoAHV-1 should be determined, as the Th1 response profile appears to be more suitable for this type of evaluation.

## Figures and Tables

**Figure 1 vaccines-12-00615-f001:**
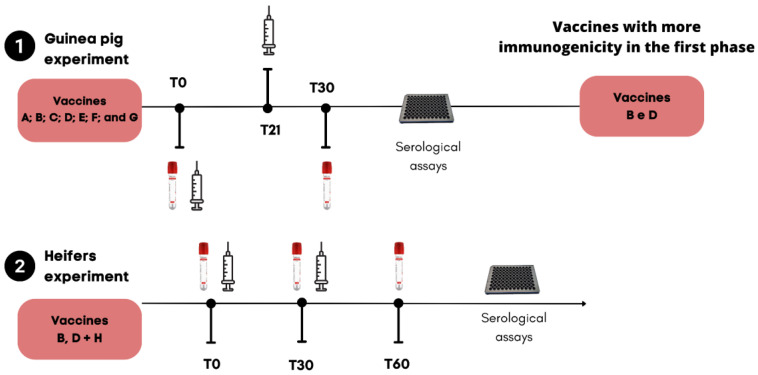
Timeline of first and second phases of vaccine inoculation in the *Guinea pig* (1) and bovine experiments (2), respectively, which were conducted to evaluate the serological immune response induced by killed vaccines and modified live vaccines (MLVs).

**Figure 2 vaccines-12-00615-f002:**
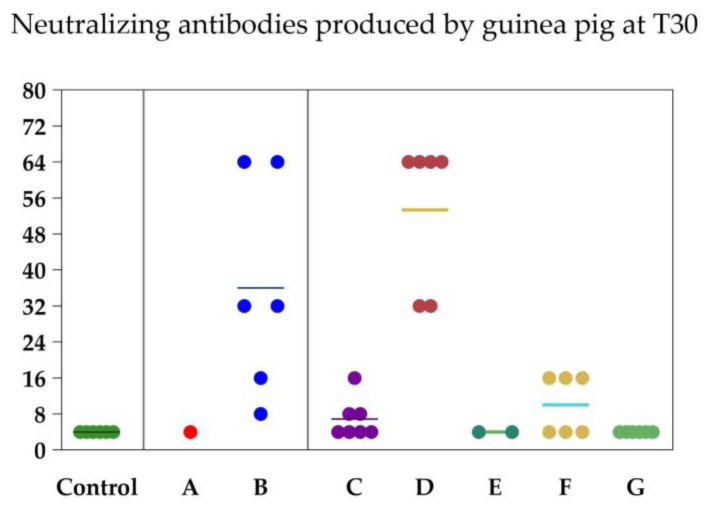
Neutralizing antibodies produced by each of the guinea pigs at T30 and by each of the cattle at T60 according to each vaccine formulation. Each circle represents an animal and each color represents a vaccine group (control, vaccine B, vaccine D and vaccine H). The marker means the average between groups.

**Figure 3 vaccines-12-00615-f003:**
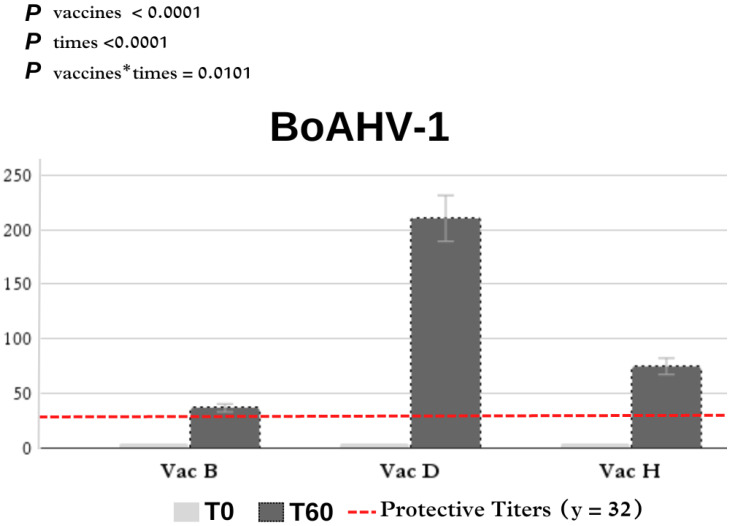
Effect of vaccination on neutralizing antibodies titers in heifers. BoAHV, bovine alphaherpesvirus; Vac, vaccine.

**Table 1 vaccines-12-00615-t001:** Effectiveness of different bovine alphaherpesvirus vaccine formulations in neutralizing the antibodies produced in guinea pig and bovine experiments.

Vaccine Formulations	Guinea Pig Experiment	Bovine Experiment
Effect of vaccines formulation	<0.0001	<0.0001
Effect of time	0.0017	<0.0001
Interaction vaccine × time	<0.0001	<0.0001
Vaccine A	4.00 ± 0.00	Reproved
Vaccine B	20.00 ± 6.66	20.00 ± 4.51
Vaccine C	5.42 ± 0.90	Reproved
Vaccine D	28.66 ± 8.10	84.76 ± 27.53
Vaccine E	4.00 ± 0.00	Reproved
Vaccine F	6.33 ± 1.34	Reproved
Vaccine G	4.00 ± 0.00	Reproved
Vaccine H	No-tested	48.74 ± 9.27
Non-vaccinated Guinea pig and bovine model	4.00 ± 0.00	

Guinea pig experiment was conducted to screen vaccines for the second experiment. Reproved—vaccines not selected for the second bovine experiment because of the low antibody titers developed after vaccination protocols; virus neutralization antibodies expressed as the mean and standard error. The effects of the vaccine group, time, and interaction between the group and time were determined using a MIXED models procedure.

**Table 2 vaccines-12-00615-t002:** Effect of different bovine alphaherpesvirus (BoAHV) vaccine formulations on the neutralizing antibodies produced by guinea pigs (T0 and T30) and bovines (T0 and T60). Capital letters in the same column demonstrate differences between vaccines; lowercase letters on the same line demonstrate differences between times. Least significant difference test.

**Vaccine**	**Guinea Pig—BoAHV-1 (Cooper)**	
	**T0**	**T30**
Non-vaccinated	4.00 ± 0.00 ^A^	4.00 ± 0.00 ^B^
A	4.00 ± 0.00 ^A^	4.00 ± 0.00 ^B^
B	4.00 ± 0.00 ^Ab^	36.0 ± 9.63 ^Aba^
C	4.00 ± 0.00 ^Ab^	6.85 ± 1.68 ^Ca^
D	4.00 ± 0.00 ^Ab^	53.33 ± 6.74 ^Aa^
E	4.00 ± 0.00 ^A^	4.00 ± 0.00 ^B^
F	4.00 ± 0.00 ^Ab^	8.66 ± 2.40 ^BCa^
G	4.00 ± 0.00 ^A^	4.00 ± 0.00 ^B^
**Vaccine**	**Bovine—BoAHV-1 (Cooper)**	
	**T0**	**T60**
Non-vaccinated	4.00 ± 0.00 ^A^	4.00 ± 0.00 ^B^
B	4.00 ± 0.00 ^Ac^	36.57 ± 7.58 ^Ba^
D	4.00 ± 0.00 ^Ac^	210.28 ± 57.09 ^Aa^
H	4.00 ± 0.00 ^Ac^	74.66 ± 14.11 ^Ba^

**Table 3 vaccines-12-00615-t003:** Effect of different bovine alphaherpesvirus (BoAHV) vaccine formulations on the total antibodies produced in the guinea pig and bovine experiments according to ELISA.

Vaccine Formulations	Guinea Pig Experiment	Bovine Experiment
Effect of vaccine formulation	0.0005	0.0005
Effect of time	0.0162	<0.0001
Interaction of vaccine × time	0.0050	<0.0001
Vaccine A	7.71 ± 7.71	Reproved
Vaccine B	533.33 ± 282.21	426.66 ± 164.86
Vaccine C	320.00 ± 96.48	Reproved
Vaccine D	693.33 ± 329.16	780.95 ± 229.19
Vaccine E	40.00 ± 40.00	Reproved
Vaccine F	70.00 ± 53.51	Reproved
Vaccine G	11.42 ± 5.01	Reproved
Vaccine H	No-tested	657.77 ± 165.01
Non-vaccinated Guinea pig and bovine model	0.00 ± 0.00	

Guinea pig experiment was conducted to screen vaccines for the second experiment. Reproved—vaccines not selected for the second bovine experiment because of the low antibody titers developed after vaccination; non-tested—modified live vaccine was not tested in guinea pig models because of the low adaptability of bovine viral diarrhea virus (BVDV) in this animal species. The effects of the vaccine group, time, and interaction between the group and time were determined using a MIXED models procedure.

**Table 4 vaccines-12-00615-t004:** ELISA in guinea pigs at days 0 and 30 following different vaccines (GLM). Capital letters in the same column indicate differences between vaccines; lowercase letters on the same line indicate differences between times. Least significant difference.

**Vaccines**	**Guinea Pig—BoAHV-1**	
	**T0**	**T30**
Non-vaccinated	0.00 ± 0.00 ^A^	0.00 ± 0.00 ^B^
A	0.00 ± 0.00 ^A^	0.00 ± 0.00 ^B^
B	0.00 ± 0.00 ^Ab^	1066.67 ± 486.47 ^Aba^
C	0.00 ± 0.00 ^Ab^	140.00 ± 103.15 ^Ca^
D	0.00 ± 0.00 ^Ab^	1386.67 ± 533.33 ^C^
E	0.00 ± 0.00 ^A^	26.66 ± 26.66 ^B^
F	0.00 ± 0.00 ^Ab^	80.00 ± 80.00 ^BC^
G	0.00 ± 0.00 ^A^	0.00 ± 0.00 ^B^
**Vaccines**	**Bovine—BoAHV-1**	
	**T0**	**T60**
Non-vaccinated	0.00 ± 0.00 ^A^	0.00 ± 0.00 ^C^
B	5.71 ± 5.71 ^A^	1188.57 ± 354.10 ^Ca^
D	5.71 ± 5.71 ^A^	2011.43 ± 354.10 ^Ca^
H	0.00 ± 0.00 ^Ac^	1440.00 ± 357.77 ^Ca^

## Data Availability

The data presented in this study will be made available on the Zenodo website.

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
