# Peer review of "Serological Responses of Guinea Pigs and Heifers to Eight Different BoAHV-1 Vaccine Formulations"

_vaccines, 2024, doi:10.3390/vaccines12060615_

Round 1
Reviewer 1 Report (Previous Reviewer 1)
Comments and Suggestions for Authors
Luana Camargo et al evaluated the serological immune response of eight different formulations of commercial vaccines against BoAHV-1.However, the mechanisms of immune protection induced by different formulations of commercial vaccines may also be different, and the efficacy of the vaccine may not be sufficient only by measuring antibody response without challenge protection test.Moreover, the time of antibody production induced by KLV and MLV is different, and it is not appropriate to measure antibodies only at 1 time point after immunization (T30 or T60).
Secondly, the presentation and writing of the manuscript are not good, and the legibility is poor.Figure 1 is missing.Bovine alphaherpesvirus 1(BoHV-1) is an irregular acronym, and also appears several times. The time descriptions of D0 and D60 in Figure 2 are different from those of T0,T30, and T60 in other tables.
Author Response
We appreciate the reviewer's comments and confirm that the points raised by the reviewer are essential. We send a response file for each point made, as well as a new version of the article with revised English.

Reviewer 2 Report (Previous Reviewer 2)
Comments and Suggestions for Authors
A review of the manuscript "Serological responses of guinea pigs and heifers to eight different BoAHV-1 vaccine formulations" demonstrated that it has improved in quality and readability. However, it still needs some corrections, which are all indicated below. The results demonstrated two distinct results: a) use of the guinea pig model for vacccine testing and b) immunogenicity of commercial vaccines.
Using an animal laboratory model for vaccine testing is an excellent advantage for the industry and academics. Vaccine immunogenicity is a problem related to several South American studies; it would be great if the authors wrote a paragraph about the possible causes because this finding highlighted the results. Several articles in South American literature address the vaccine-low immunogenicity problem. The authors can read them and gain some insights, and the citation of these references should be up to their convenience.
Line 12 – dairy and beef cattle
Line 29 – Alphaherpesviridae – italic
Line 29 and 37 – remove "type. "Bovine alphaherpesvirus 1"
Line 40 -dormant state – latent state
Line 47 – beef and dairy (correct) – different from line 12
Line 57 - without technical training - without serological testing
Line 112 – punctuation
Line 118 – punctuation
Lines 108 to 125 – It is suggested to reduce it. All these explanations are not necessary. The authors can just cite an article. A search in South American literature was possible to find the (Ciencia Rural 37:(4), 2007. Guinea pigs as a model test of bovine herpesvirus type 1 and bovine viral diarrhea virus inactivated vaccines. https://doi.org/10.1590/S0103-84782007000400023).
Line 142 – 144 – Figure 1?
Line 146 – guinea pig and bovine samples – lowercase
Line 157 – Highest
Line 163 and 169 – 109 and 105 (should be superscript)
Lines 163, 169, and 176 – "405" underwritten
Line 188 – guinea pig model
Comments on the Quality of English Language
Minor editing of English language required
Author Response
We appreciate the reviewer's comments. We submit the article for a thorough and intensive English review. We believe the reviewer's comment is very important and that is why we added a paragraph about the use of Guinea Pigs and vaccine immunogenicity. All corrections were answered in the attached pdf document and all corrections were highlighted in yellow in the new version of the article sent.

Reviewer 3 Report (Previous Reviewer 3)
Comments and Suggestions for Authors
The manuscript has been improved. For the benefit of the reader, however, a number of points need clarifying.
1. Gibbs and Rweyemanu reported that BoHV-1 does not infect guinea pigs (1977). The authors should cite this paper and discuss why vaccines B and D induced antibodies in the animals.
2. BoAH V-1 and BoHV-1 are mixed throughout the manuscript.
3. Strain name and serovar name are mixed (lines 284-289).
Comments on the Quality of English LanguageModerate editing of English language required.
Author Response
We appreciate the reviewer's comments. We submit the article for a thorough and intensive English review. We also add that we believe the reviewer's comments were extremely important. The changes made are highlighted in yellow in the new version sent and described below.

Round 2
Reviewer 1 Report (Previous Reviewer 1)
Comments and Suggestions for Authors
I generally agree with the authors' revisions to the manuscript, but the format of the results is simplistic and almost tabular. Could the proposed experimental results be presented in a more diverse format to increase readability?
Author Response
We appreciate the review and have inserted our response in the attached PDF document.

This manuscript is a resubmission of an earlier submission. The following is a list of the peer review reports and author responses from that submission.
Round 1
Reviewer 1 Report
Comments and Suggestions for Authors
BoHV-1 is associated with respiratory and reproductive diseases, causing huge economic losses to the cattle industry. Vaccination is one of the effective ways to control and prevent BoHV-1 infection and transmission. Luana Camargo et al. evaluated the serological immune response against BoHV-1 induced by eight different formulations of commercial vaccines have only confirmed that MLV can induce neutralizing antibodies at a certain level of protection, but no cellular immune indexes have been determined and no animal challenge protection tests have been conducted, which is insufficient to reflect the effectiveness of the vaccine.
Major comments:
1. Table A1. The information of various commercial vaccines is unknown. The strains and types of vaccines are unknown. Vaccine antigen content or virus titer of live vaccine, etc
2. Table A1. The immune doses and pathways of various commercial vaccines are somewhat different. However Fig 1 "The animals were vaccinated subcutaneously with two doses of the commercial formulations at an interval of 21 ".Such an immunization program is not the optimal immunization strategy for various commercial vaccines and does not reflect the immune effect of vaccines well.
3.Table 4 Vac. B\D\H The differences in the induced neutralizing antibodies of these three vaccines may be caused by the difference in vaccine antigen content or virus titer of these vaccines, or by incorrect immunization procedures? For example, Vac.B recommended immune doses and procedures: Administer one dose of 2 ml by intramuscular injection in the neck muscles? In Fig 1, why is immunization performed twice?
4. 2.3.2 ELISA. The use of 50 μL MDBK cells and 50 μL BoHV-1 as coated antigens may affect the specificity and sensitivity of this ELISA method, and commercial kits are recommended.
Minor comments:
1. Line 20 “ a competitive enzyme-linked imunosorbent assay (ELISA)”
However, Line 110 and Line 136 "A direct ELISA", please check;
2. Line 96 Table 1, but Table 1 is not found in the document? Is it Table A1?
3. Line 132-134 “Antibody titers were considered the reciprocal of the higher serum dilution that prevented the production of dense bodies. Antibody measurements”. This sentence is somewhat confusing;
4. Figure 2. What was the protective antibodies titers? neutralizing antibodies titers?
5. Table 2 and Table 5 lack statistical analysis;
6. Line 186-187 "five of six animals died from vaccine A, and two of six from vaccine C." What's the cause of animal death? Too high an immune dose?
7. The references are not new enough, with only one in the past three years.
Author Response
We thank the reviewer for his/her comments. We are sending you the answers to all the reviewers' comments and questions and the manuscript with the requested changes in pdf format.

Reviewer 2 Report
Comments and Suggestions for Authors
The authors vaccinated guinea pigs and heifers to evaluate the serological response (immunogenicity) of commercial BoHV-1 vaccines. Antibody levels were assessed by ELISA and virus-serum neutralization, which allowed them to evaluate the total serological response and the levels of neutralizing antibodies. This type of study is highly valid, and several groups of researchers have been concerned about the quality of commercial vaccines. This manuscript has several weaknesses that prevent it from being published. Some of the observations are highlighted below.
- The correct and current way of writing the name of the virus is bovine alphaherpesvirus 1 (BoAHV-1) or bovine alphaherpesvirus 5 (BoAHV-5), and should be standardized throughout the text. The ICTV recommends another more up-to-date form, "Varicellovirus bovinealpha1". Taxonomy has become very dynamic recently, and it's a good idea to consult with the ICTV website.
- Another option would be to mention the most up-to-date nomenclature (formally bovine herpesvirus 1 - BoHV-1).
Introduction
- correct the description of BoHV-1, family, genus.
- at the end of the first paragraph, include the description of BoHv-5, as this virus is important in South America.
- BoAHV-1 is a problem for beef and dairy cattle farming.
- "BoHV-1 persists in herds due to silent carriers with the characteristic, subclinical disease status, which is thus difficult to diagnose [1]"—confusing sentence. Detecting latently infected animals is not difficult; a simple serological test can do the trick.
- Similar to most viral infections, immune responses against BoHV-1 can be divided into preventive (???) (humoral) and cellular and humoral responses that aid recovery. It needs to be a more explicit and more appropriate sentence.
- Although neutralizing 54 antibodies can eliminate free BoHV-1, disease recovery primarily depends on cell-mediated 55 immunity—another confusing sentence.
- BoHV vaccination aims to reduce viral excretion, prevent clinical manifestations, and reduce viral circulation in the herd. MLVs are not recommended for vaccinating pregnant animals.
- vaccine tests are extensive in all parts of the world. Remove the sentence referring to South America.
- a near-perfect - perfect or imperfect. Replace with high association, high ratio...
M&M
- The serological responses of guinea pigs and heifers against 90 BoHV-1 induced by commercial vaccines were evaluated in two phases—nonsense sentence.
- The guinea pig experiment (experiment 1) was conducted according to the CAMEVET protocol (????), adopted by Argentina (???)- needs to be supplemented. ... for the evaluation of the efficacy of the vaccine (the authors evaluated the immunogenicity of the vaccines)
- Antibody titers were considered the reciprocal of the higher serum dilution that prevented the production of cytopathic effect (CPE). Dense bodies?
Figure 1 - Best vaccine = more immunogenicity. Correct legends (1 and 2?) do not appear in the figure. I suggest dividing the diagram—one line for the guinea pig experiment and another for the cattle experiment.
- ELISA - is this protocol complete? It needs to be better described.
- Serological tests - describe in the material and methods the samples tested in each test. Guinea-pig x ELISA? I think this needs to be clarified.
- Statistical analysis - which serological results were analyzed. ELISA or VNT values?
Results
- Check the numbering of the tables.
- Table 3 and Table 4 - the authors could merge the two tables. There is no need to compare zero-day serology with post-vaccination serology.
- Toxic effect of vaccines A and C - how was it proven?
- The authors used guinea pigs and cattle to assess the immunogenicity of BoHV-1 vaccines. To do this, they carried out an initial evaluation of the experimental model and then vaccinated the target species. The lowest serological dilution tested was 1:4, so the authors consider that any titer below four would be negative.
- Holstein × Giro or Girolando????
- The BoHV-1 Ab response to vaccination in guinea pigs, as measured by ELISA and VN, - needs to be described in the M&M how the ELISA for guinea pigs was carried out.
- These results indicated that the developed guinea pig model is a 239 novel and reliable tool for estimating vaccine potency and predicting the efficacy of cattle vaccines. - Novel?
- The minimum antibody titers needed to protect animals from the BoHV-1 challenge suggested by Pospísil et al. (1996) [14] are ≥ 1:16 or 1:32 - No. 1:16 or 1:32 are de dilution, titer must be referred to as 16 or 32.
In the first study, KLV A, C, E, F, and G did not induce satisfactory - and the vaccine A and C toxic effect?
- What could be the causes of the low immunogenicity of the vaccines? See literature
- The authors did not discuss the fundamental points of vaccines: duration of the immune response? Revaccination is recommended every year.
Comments on the Quality of English LanguageThe work requires a thorough revision of the English language
Author Response

(The authors gave the same response as above.)

Reviewer 3 Report
Comments and Suggestions for Authors
The authors demonstrated that a live ts mutant virus vaccine could induce the most effective immune response against BoHV-1 in cattle. This finding is of considerable interest. However, they also indicated that MLV B and D induced stronger response than all the KLVs in guinea pig. Did vaccines B and D replicate in guinea pig? If the animal is susceptible to BoHV-1, the authors should perform challenge experiment after vaccination. The authors should also revise ELISA procedure completely. There is no description about guinea pig experiment. Did they coat the plate with MDBK cells? What was the substrate?
Minor points:
1. Line 2. Delete immune. The title should be concise.
2. Line 11. Bovine herpesvirus.
3. Line 20 Delete competitive.
4. Line 24. Neutralizing not protective.
5. Lines 38-40. This sentence should be revised.
6. Lines 60-62. This sentence should be revised.
7. Lines 70 and 307. MLV.
8. Line 109. Virus neutralization (VN).
9. Line 112. Delete virus neutralization ( ).
10. Line 134. Delete antibody measurements.
11. Line 170 and others. Table No. should start from 1.
12. Tables 2 and 5 (original No.). Delete BoHV-1. Combine non-vaccinated.
13. Line 178. Delete animals.
14. Tables 3, 4, 6, and 7. Delete vac.
16. Line 235. Immune or antibody not Ab.
17. Lines 257 and 271. BoHV-1.
Comments on the Quality of English LanguageModerate editing of English language required.
Author Response

(The authors gave the same response as above.)
